# Preparation and Mechanism Analysis of Boiling Resistance of the Fresh Alum-Free Sweet Potato Vermicelli Containing Gliadin Fractions

**DOI:** 10.3390/foods14010081

**Published:** 2025-01-01

**Authors:** Tingting Liu, Zhifang Men, Changjiangsheng Lai, Xijun Lian

**Affiliations:** 1Tianjin Key Laboratory of Food Biotechnology, Institute of Collaborative Innovation in Great Health, College of Biotechnology and Food Science, Tianjin University of Commerce, Tianjin 300134, China; 120220371@stu.tjcu.edu.cn (T.L.); lianxijun@tjcu.edu.cn (X.L.); 2School of Foreign Languages, Tianjin University of Commerce, Tianjin 300134, China; grace.men@tjcu.edu.cn

**Keywords:** sweet potato vermicelli, ω-gliadin, αβγ-gliadin, boiling resistance

## Abstract

Alum, an essential additive in sweet potato vermicelli (SPV) production, is harmful to health. To eliminate the harm to the human body caused by alum in sweet potato vermicelli, and considering the different viscous properties of gliadin fractions, an experiment was performed to replace alum with gliadin fractions to enhance the boiling resistance of SPV in this study. The results showed that the longest boiling-resistant time of fresh SPV extended to 34.31 min when swelling the dough binder at 50 °C for 5 h, adding a 2% complex of ω-gliadin + αβγ-gliadin at a ratio of 1:1, and mixing at 70 °C for 20 min. The result was 95.7% higher than in the control. Starch swelling and freeze–thaw processes could partially replace the role of alum in preparing SPV. The results of FTIR and *^13^C* solid-state NMR showed that the esterification reaction of ω-gliadin and αβγ-gliadin and hydrogen bonds between sweet potato starch and gliadin fractions reinforced the boiling resistance of vermicelli. There was no ordered area of starch in the new water-resistant vermicular. The gliadin fractions formed crystal with a diffraction angle of 17.38° (3.25 Å). Long-term cold storage could improve the boiling resistance of fresh sweet potato vermicelli. Additionally, the short-term retrogradation of sweet potato amylose significantly reduces its boiling resistance. The study provides new primary data and theoretical support for the industrial application of alum-free fresh sweet potato vermicelli.

## 1. Introduction

Sweet potato vermicelli (SPV) is one of the important staple foods of Chinese people; it is a local characteristic snack in the Jiangxi, Guangxi, Shaanxi, Hunan, Fujian, Sichuan, and Guizhou provinces in China. During the development of SPV over thousands of years, the Chinese invented the technology of using alum to improve its quality. According to Jiang et al. [1], the alum contents of three domestic brands of sweet potato vermicelli were as follows: Tianyu had 106.7 mg/kg, Shenglin had 48.3 mg/Kg and Longxu had 42.1 mg/Kg. Alum improves the quality of vermicelli by reducing amylose leakage and forming ionic bonds [2]. However, an increasing number of experimental results have shown that the aluminum present in alum is harmful to the human digestive tract and nervous system [3,4]. Many Chinese consumers of sweet potato vermicelli have experienced symptoms such as stomach bloating and cerebellar atrophy, confirming the results of these scientific experiments.

Boiling resistance is one of the main characteristics affecting the application of vermicelli. It is an essential index of vermicelli quality [5], especially when vermicelli is used as a base material of the hot pot in China. The phenomenon frequently observed in hot pot cuisine is that the SPV becomes adhesive and develops cracks after water absorption, rendering it difficult to handle with chopsticks. It eventually completely dissolves in the soup, resulting in increased viscosity in the soup and a bad taste. Gull et al. showed that increased cooking loss can be attributed to the lack of gluten in these flours [5]. The addition of gluten will improve the resistance of vermicelli to boiling. Strong resistance to boiling is the main characteristic of SPV following the addition of alum. Researchers have tried many methods to replace the alum in SPV. Feng et al. reported that the addition of 0.5% chitosan, 0.5% sodium alginate, 0.5% xanthan, 3.0% gluten, and 3.0% egg white protein extended the cooking break time of SPV from 4.5 min to 28.5, 30.5, 27.5, 23.5, and 25.5 min, respectively [6]. Those times were similar to the breaking time of 27.0 min achieved by adding alum [6]. Additionally, SPV’s cooking break time increased significantly to 48.5 min with the addition of 0.5% chitosan, 0.5% sodium alginate, and 3.0% egg white protein complex. These additives induced the formation of a less crystalline structure in the wet SPV [7]. Adding polysaccharides and egg white protein to SPV dramatically affects its taste, hindering its industrial application. This is because these substances are not grain components.

Gluten, which is a component of flour, is made up of 65% gliadins (α-, β-, γ- and ω-gliadins) and 35% glutenin [8,9]. The water-insoluble nature of gluten protein and its poor dispersibility in vermicelli restricts its widespread use in this food product. The viscous and elastic behaviors of gluten are induced by gliadin and glutenin, respectively. Gliadin is a single-chain protein linked by hydrogen bonds and hydrophobic bonds, which have low bond energy and are easily “broken”. As such, gliadin makes gluten sticky and stretchable [10]. The viscosity of gliadins may improve the boiling resistance of SPV. According to our latest research [11], ω-gliadin fraction (9.67% of gluten) and αβγ-gliadin (12.43% of gluten) could be prepared from gluten powder in batches by simple refrigeration and freeze–thaw methods. Then, the two gliadin fractions (ω-gliadin and αβγ-gliadin) are added in dough to investigate their effects on the boiling resistance of SPV in this paper. The impact pattern of ω-gliadin/αβγ-gliadin addition on the boiling resistance of SPV was deduced based on the results of optical micrographs, disulfide bond contents, IR, *^13^C* solid-state NMR, X-ray diffraction, and DSC. This paper presents the essential characteristics of water-resistant SPV, including the minimal crystal morphology of granules in vermicelli. The objective of this experiment is to investigate the feasibility of utilizing gluten protein as a substitute for alum to extend the cooking time of sweet potato vermicelli, thereby enhancing both the safety and quality of the product.

## 2. Materials and Methods

### 2.1. Materials

Sweet potato starch (food grade, the production date was 12 March 2024) was purchased from Shandong Yusheng Food Co., Ltd., Shandong, China. Gluten (Purity 99%) was purchased from Henan Midan ’er Trading Co., Ltd., Henan, China. Other chemical reagents were purchased from Tianjin Fuyu Fine Chemical Co., Ltd., Tianjin, China. Henan Xinheyang Alcohol Co., Ltd., Henan, China supplied ethanol. At the same time, Beijing Solarbio Technology Co., Ltd., Beijing, China provided Tris, glycine, disodium EDTA, urea, and DTNB reagents. High-temperature α-amylase (12,000 U·mL^−1^, from Bacillus licheniformis) was purchased from Beijing Solarbio Science & Technology Co., Ltd., Beijing, China. The experiments were conducted from March to June 2024.

### 2.2. The Processing Technology of Vermicelli

#### 2.2.1. Pre-Treatment of Sweet Potato Starch for Vermicelli Thickening

The pre-treatment of sweet potato starch for vermicelli thickening was prepared according to our reference [12]. Sweet potato starch at a set weight (100 g) was added to 1000 mL of distilled water and gelatinized by continuous stirring for 3 h at 45 °C until it became viscous. The resulting gel was frozen at −18 °C overnight and thawed at room temperature. Subsequently, it was centrifuged at 3500× *g* for 3 min and washed three times with distilled water to remove impurities. The wet swollen sweet potato starch was then used as a dough binder for the vermicelli.

#### 2.2.2. Preparation and Determination of Cooking Break Time of Vermicelli

The wet SPV was prepared in accordance with the methodology described in the literature [6]. We added the gliadin fractions to the wet swollen sweet potato starch, stirred with a mixer at 200 r/min at different temperatures and times, and mixed evenly. The mixture was poured into the pot and heated on an electromagnetic stove (Power 300 W) until it became completely gelatinized. The dough was prepared by adding 80 g of sweet potato starch and gliadin fractions, followed by kneading. The vermicelli was then pressed using a vermicelli press. Ten vermicelli samples were heated in a boiling water bath in an induction cooker filled with 1 L of water. When cooking, the cooker was set to 1000 w and kept slightly boiling. The breaking time of the first vermicelli was observed and recorded. The shorter the breaking time, the worse the cooking resistance.

#### 2.2.3. Preparation of Gliadin Fractions

The gliadin fractions were prepared from gluten following the method in the literature, albeit with minor modifications [11,13]. A mixture of 200 g of gluten flour and 6 L of 65% ethanol was stirred at 35 °C for 3.5 h. The solution was centrifuged at 3500× *g* for 3 min to obtain a clear supernatant. After being stored at 4 °C for 24 h, the first sediment (gliadin 1) appeared in the supernatant and was separated by pouring out the supernatant. The main components of the mixture were ω-gliadin and alcohol-soluble glutenin (with molecular of 40,397/17,837 g/mol). The remaining supernatant was frozen at −18 °C for 24 h and thawed at room temperature. The second sediment, gliadin 3, was isolated through centrifugation at 4000× *g* for 5 min. Its main component was αβγ-gliadin (with a mean molecular weight of 31,925). The wet gliadin fractions were added directly to the vermicelli and their contents were calculated using dry weight.

#### 2.2.4. The Effects of Swollen Temperature and Time on Cooking Break Time of SPV

The aqueous starch solution was allowed to swell at temperatures of 30 °C, 35 °C, 40 °C, 50 °C, and 60 °C for 2, 3, 4, 5, and 6 h, respectively. The starches obtained were utilized as a dough binder to prepare SPV. The optimal swelling temperature and time were determined based on the cooking break time of SPV. The wet SPV was subsequently dried to a constant weight in an oven at 60 °C and then ground using 100-mesh sieves for the purpose of performing IR spectroscopy, 13C solid-state NMR, X-ray diffraction, and DSC.

#### 2.2.5. The Effect of Adding Gliadin 1, Gliadin 3 or a Combination of Both on the Cooking Break Time of SPV

The gliadin 1 (0%, 0.1%, 0.5%, 0.8%, 1.0% of dough binder)/gliadin 3 (0%, 0.5%, 1.0%, 2.0%, 4.0% of dough binder)/gliadin 1 + gliadin 3 (ratio of 1:1, 0%, 0.5%, 1.0%, 2.0%, 4.0% of dough binder) were blended with dough binder at 30 °C, 40 °C, 50 °C, 60 °C, and 70 °C for 20 min, 40 min and 60 min, respectively. The complexes were blended with sweet potato starch to prepare SPV. The resulting wet SPV was boiled in water to determine the time it took to break apart.

#### 2.2.6. The Effects of Cold Storage on the Retrogradation Rate of SPV Containing Gliadin Fractions

The wet SPV samples were restored at 4 °C for 1, 2, 3, 4, 5, 6, 7, 8, 9 and 10 d, respectively, and their retrogradation rates were determined according to our reference [14]. First, 10 pieces of extruded noodles with similar lengths and thicknesses were selected. They were boiled in a pot of boiling water for 3 min; then, they were transferred to cold water for cooling. Once cooled, they were put in a 4 °C refrigerator for retrogradation at intervals of 1 to 10 days. After retrogradation, these noodles were cut into segments and dried in a 50 °C oven to determine the dry weight (M1). The dried noodles were crushed and redissolved in deionized water. The high-temperature amylase was added to hydrolyze not-retrograded starch at 90 °C for 2 h while we stirred intermittently. Then, the solutions were centrifuged at 3500× *g* for 5 min to obtain precipitates. After, the precipitates were washed with distilled water three times and dried to a constant weight (M2). The retrogradation rate = ((M1 − M2)/M1) ×100%.

### 2.3. Light Microscopy

The wet SPV was dispersed at a weight of 0.1 g in 10 mL of NaOH (2 mol/L) and a drop of the solution was spread well on a microscope slide. After drying, samples placed on a carrier table for observation. Photomicrographs were taken with an optical microscope (CKX53; Olympus Co., Tokyo, Japan). The brightness and magnification were adjusted accordingly [15].

### 2.4. FTIR Spectroscopy

All samples were dried in an oven at 60 °C to a constant weight before being blended with spectroscopic-grade KBr. After being pressed into flakes, the samples were placed on a slide holder to record the spectral curves at 27 °C using a Fourier transform infrared spectrometer (Perkin-Elmer, Buckinghamshire, UK). The different secondary structures of the gliadin fractions correspond to the infrared absorption range: α-helices, 1670–1680 cm^−1^; intermolecular β-sheets, 1612–1620 cm^−1^/1680–1695 cm^−1^; intramolecular β-sheet, 1625–1642 cm^−1^; β-turns, 1650–1660 cm^−1^; random coil, 1642–1650 cm^−1^. The secondary structure content of the gliadin fractions was calculated using Peakfit software (PeakFit v4.12, SeaSolve Software Inc., San Jose, CA, USA; Framingham, MA, USA) [16].

### 2.5. Determination of 13C Solid-State NMR Spectroscopy

The dried and powdery SPV were loaded into a 5 mm rotor with a 13C frequency of 150.87 kHz, corresponding to a 90° pulse width of 2.4 μs at room temperature. A JEOL ECZ600R 600 MHz spectrometer (JEOL RESONANCE Inc., Tokyo, Japan) was used to record the 13C solid-state NMR signals and transform signals into data.

### 2.6. Determination of X-Ray Powder Diffraction (XRD)

A D/MAX-2500 Advance diffractometer (Rigaku, Tokyo, Japan) was used to determine the X-ray diffraction patterns of powdery SPV. It worked at 200 mA and 40 kV. The diffraction angle (2θ) was scanned in a range of 3°~60° with a step size of 0.02°. The counting time was 0.8 s.

### 2.7. Determination of Differential Scanning Calorimetry (DSC)

DSC was carried out on a DSC404C from Netzsch Instruments NA LLC (Burlington, MA, USA). The powdery SPV samples, at the weight of 5.0 mg, were placed in aluminum pans and heated from 25 °C to 200 °C with a heating rate of 1 °C/min [14].

### 2.8. Statistical Analysis

Data were expressed as mean ± standard deviation and all tests were performed in triplicate. Significant differences/extremely significant differences between control and experimental groups were compared via Dunnett’s test (*p* < 0.05/0.01) [17].

## 3. Results and Discussion

### 3.1. Effect of Swelling Temperature and Time on the Resistance of Vermicelli to Boiling

We explored a novel and effective approach to substituting alum’s role in reducing amylose leakage and forming ionic bonds in SPV. The fundamental principle of the new method involves damaging the granules in the SPV dough binder through swelling, freezing, and thawing. This process releases more amylopectin, enhancing the vermicelli’s resistance to boiling. The ice matrix exerts pressure at freezing temperatures, compromising the granules. During this process, the moisture absorbed by the starch granules freezes, causing damage to the granules due to the pressure exerted by the increased ice matrix and the leaching of amylopectin after thawing.

It is known that the gelation temperature range of sweet potato starch is 55 °C to 85 °C [18], and so we set the swelling temperature (sub-gelatinization temperature) to run from 30 °C to 60 °C. Table 1 shows the effect of swelling temperature and time on the boiling resistance of vermicelli. Swelling temperature and time have a significant influence on the boiling break time of SPV. When the swelling temperature is lower than 35 °C, the SPV exhibits poor formability. This may be attributed to the reduction in water permeating the granules and the absence of sufficiently large ice crystals. These large ice crystals are necessary to break the granules and leach out the amylopectin, and a lack of them leads to the poor formability observed.

When the swelling temperature exceeds 40 °C, the breaking time of vermicelli in boiling water obviously becomes longer. That is, the increase in swelling temperature prolongs the break time. The worst and most favorable conditions are swelling at 40 °C for 2 h and 50 °C for 5 h, respectively. The breaking times in boiling water in Table 1 reach 6.85 min and 15.84 min, respectively. When the swelling temperature is increased to 60 °C, the amylose in large granules may leach out and gelatinize on the surface of other granules to hinder water absorption. The crystals in the granules will then reduce the breaking time in the boiling water of SPV. If only heat treatment at the sub-gelatinization temperature is carried out, without freeze–thaw processing, sweet potato starch granules merely become rough on the surface [19]. As a result, it may not be possible to produce sweet potato vermicelli without alum. Although the boiling time of vermicelli after swelling treatment is obviously prolonged, the boiling time of vermicelli is still shorter than that of vermicelli with alum, standing at 28 min [7]. A small amount of gliadin fractions is added to prolong the time it takes for SPV to break when boiled in water.

### 3.2. The Effects of Adding Gliadin Fractions on SPV Cooking Time

Table 2 shows the effect of the amount of ω-gliadin added, the mixing temperature, and time on the breaking time when boiling water containing SPV. When ω-gliadin is added in the range of 0.5~0.8%, the SPV break time in boiling water is significantly prolonged. Mixing temperature and time interact in the preparation of SPV; low-temperature treatment for a long time and high-temperature treatment for a short time are beneficial in terms of prolonging the break time in boiling SPV water. The optimum conditions are 0.1% of ω-gliadin and mixing at 70 °C for 40 min. The breaking time in boiling water for this sample is 25.00 min (marked in red), which is 40.6% higher than that of the corresponding blank group. This is 17.78 min, as shown in Table 2.

Table 3 shows the effect of the amount of αβγ-gliadin addition, mixing temperature, and mixing time on the boiling break time of SPV. It is noted that the boiling break time of samples with 1.0~2.0% αβγ-gliadin addition is relatively longer. The longest one is seen in the conditions of 2.0% αβγ-gliadin addition, with mixing at 40 °C for 60 min. The break time in boiling water for this sample is 29.53 min (marked in red), which is 59.2% higher than that of the corresponding blank group, 18.55 min, as shown in Table 2 (marked in green).

Table 4 shows the effect of the amount of ω-gliadin and αβγ-gliadin when added at a ratio of 1:1, the mixing temperature, and the mixing time on the boiling break time of SPV. It is noteworthy that there are more samples with break times longer than 30 min in the group of 2.0% ω-gliadin + αβγ-gliadin addition. The longest break time is seen in the conditions of 2.0% addition and mixing at 70 °C for 20 min. The break time in boiling water for this sample is 34.31 min (marked in red), which is 69.4% higher than that of the shortest, 20.25 min, as seen in Table 4 (marked in green). ω-gliadins are sulfur-poor, while αβγ-gliadins are sulfur-rich. The former is more hydrophilic compared to the latter, while the latter is more hydrophobic compared to the former [20]. Moreover, the former can serve as an emulsifier for the latter, interacting with sweet potato starch upon the mixture of the two. During the boiling process, gliadin polymerization dramatically elevates the viscoelasticity of SPV [21], resulting in the extended boiling break time of SPV. During this process, the formation of disulfide bonds in αβγ-globulins may contribute to the stabilization of the mixture’s structure. Typically, the term “alcohol-soluble proteins” (gliadin in the paper) refers to the mixed proteins found in the ethanol extract of gluten powder, primarily consisting of α, β, γ, and ω-gliadins, along with a minor amount of alcohol-soluble glutenin. These components exist in the form of polymers, and during the purification process, the aggregation of these polymers increases, leading to reduced hydrophilicity and difficulty interacting with starch. Therefore, it is essential to perform component separation.

Table 5 shows the effect of cold storage time on the resistance of different vermicelli to boiling. Compared to fresh and wet vermicelli, the boiling break time of all vermicelli samples is significantly decreased after cold storage for 1d, which is attributed to the amylose retrogradation. With increasing cold storage time, the cooking break times of all samples are significantly increased, which is related to the retrogradation of amylopectin during long-term cold storage. The best result is seen for composite gliadin vermicelli 1 (1.0% addition of ω-gliadin + αβγ-gliadin (1:1) and being mixed at 40 °C for 20 min), and its boiling break time reaches 40.6 min. Cold storage for more than 5 days before drying improves the boiling resistance of SPV in the dry state.

To investigate the mechanism behind the extended cooking time of SPV due to the addition of gliadin fractions, samples with the shortest and longest break times are dried and ground for detailed analysis. This includes optical micrographs, disulfide bond content, IR, 13C solid-state NMR, X-ray diffraction, and DSC.

### 3.3. The Optical Micrographs of Samples with Different Boiling Break Time of SPV

Figure 1 shows the optical micrographs of samples with different boiling break times used for SPV. For the swelling samples with the shortest and longest break times, shown in Figure 1a,b, it is evident that the aggregates of amylose and amylopectin, shown in Figure 1a, have longer lateral branch lengths. This is favorable for crystal formation. Crystal is a major factor in reducing the boiling break time of SPV. These longer lateral branch lengths in Figure 1b might break into shorter ones during ice crystal growth. For the ω-gliadin addition group in Figure 1d and its control sample in Figure 1c, the appearance of many spicules in the former suggests that the aggregation of the ω-gliadin + amylopectin complex is present. This aggregation prolongs the boiling break time of SPV. For the αβγ-gliadin addition group in Figure 1f and its control sample in Figure 1e, the aggregation of the αβγ-gliadin + amylopectin complex in Figure 1f presents a gel-like structure. The combination of αβγ-gliadin with amylopectin hinders the evaporation of moisture bound to amylopectin, resulting in less retrogradation of amylopectin and extending the boiling break time of SPV. The aggregation volume decreases for the complex group in Figure 1h and its control sample in Figure 1g, and a low degree of gelation is present. Hydrophilic ω-gliadin acts as an emulsifier in the complex and more water molecules can be volatilized in SPV. The more stable structure of the αβγ-gliadin + ω-gliadin + amylopectin complex prolongs the cooking time of the vermicelli. The aggregates of ω-gliadin and αβγ-gliadin in Figure 1i and j present filaments and ribbons, respectively.

### 3.4. FT-IR Spectra of Samples with Different Boiling Break Time of SPV

Figure 2 shows the FTIR spectra of samples with different SPV boiling break times. Working according to reference [9,10,22,23,24], all the infrared absorption peaks in Figure 2 are assigned as follows: ~3439 cm^−1^ for the O–H stretching vibration of sweet potato starch or N–H stretching vibration of gliadin fractions; ~2925 cm^−1^ for the C–H stretching vibration of methylene; ~1651 cm^−1^ for the C=O stretching vibration of amide I band; ~1636 cm^−1^ for the H–O–H stretching vibration of water; ~1535/1541 cm^−1^ for the C–N stretching vibration and N–H bending vibration (amide II band); ~1449 cm^−1^ for C–N bending vibration of the amide III band; ~1269 cm^−1^ for H–N–C bending vibrations (amide III band); and ~1081/1020 cm^−1^ for C-O-C stretching and CO (-COH) stretching vibration (ordered/amorphous regions of starch).

Comparing the results in Figure 2a and Figure 2b, the peak for the O-H stretching vibration of SPV sample with shorter boiling break time shifts from 3439.4 cm^−1^, shown in Figure 2a, to 3442.7 cm^−1^, which is the sample with the longer boiling break time, as shown in Figure 2b. The infrared absorption band shifts towards a higher wavenumber, indicating that the amount of amylopectin leached out by the swelling of the granules in the SPV is increasing, as the peak of amylopectin in this range is at a higher wavenumber. A high content of amylopectin is beneficial for improving the boiling resistance of the vermicelli. The infrared absorption of the water, shown in Figure 2b, also moves to a higher wavenumber, which may be related to the relatively high content of the combined water in the sample. When the sample is heated at 70 °C in water for 40 min (Figure 2c), the absorption peak of water molecules returns to 1636.7 cm^−1^, indicating that some amylopectin in the sample may be broken down into amylose during the heating process. When compared with Figure 2c,d,i, it can be observed that the absorption of the amideⅠbond, which is initially at 1658.9 cm^−1^ in Figure 2i, shifts to a lower wavenumber of 1651.2 cm^−1^ in Figure 2d after ω-gliadin is added to SPV samples. This shift indicates that a hydrogen bond may be formed between the carbonyl group within the amide bond of ω-gliadin and the hydroxyl group of starch. This deduction can also be verified by the low-field shift of the O-H stretching vibration of the sample. However, the absorption peak of the amide Ⅱ bond at 1535.1 cm^−1^ is lost in Figure 2d, suggesting that the structural changes of ω-gliadin from an intra-molecular β-sheet to an α-helix, shown in Table 6, hinder these vibrations.

In Figure 2d, the presence of a sharp peak at 1269.4 cm^−1^ might be ascribed to the fact that the helical structure of ω-gliadin combines with sweet potato amylose, thereby forming the crystal. This combination leads to the appearance of the specific sharp peak observed in the figure. Figure 2e exhibits an absorption peak at 2973.6 cm^−1^. This peak is probably the result of the polymerization of aromatic amino acids present in a small quantity of protein bound to the sweet potato granules [22]. Moreover, it is noted that this reaction has the potential to take place at a temperature of 40 °C and last for 60 min. This finding provides a direct basis for the light gray appearance of SPV.

The appearance of the peaks within the range of 1080 to 1091 cm^−1^ in Figure 2a–e implies that the crystalline region is present in the unbroken small granules of these samples [23]. This is different from the absorption at 1047 cm^−1^, which is typically observed in normal granules for the same region [25]. The presence of this crystalline region will reduce the cooking time of vermicelli. After mixing with αβγ-gliadin (Figure 2f), the O-H stretching vibration of SPV moves to a higher wavenumber (from 3419.6 cm^−1^ in Figure 2e to 3432.1 cm^−1^ in Figure 2f), indicating that the amount of intermolecular hydrogen bonds in SPV decreases. The appearance of the absorption peak at 1020.6 cm^−1^ indicates that the crystalline region of SPV is transformed into an amorphous region after the interaction between αβγ-gliadin and sweet potato starch [25]. SPV with a high content of the amorphous regions is more resistant to boiling water. The absorption peaks of all the amide bonds in this sample disappear. When combined with the secondary structure analysis of αβγ-gliadin in Table 6, it can be inferred that the β-sheet structure of SPV may overlap with these absorptions. When the ω-gliadin + αβγ-gliadin (1:1) mixture is added to SPV (Figure 2g,h), the absorption peak for the amideⅠbond reappears. However, there remain no absorption peaks for the amide Ⅱ bond. New absorption peaks at 1269.1 cm^−1^ and 1250.1 cm^−1^ appear simultaneously in these two samples. The former refers to the C-O bond of the ester groups, and the latter represents the N-H group of the amine-III bond [26]. The intensity of these two peaks grows as the amount of gliadin fractions increases. Moreover, sweet potato vermicelli (SPV) displays a greater resistance to water boiling. Given that ω-gliadin contains high levels of Ser and Tyr, and that αβγ-gliadin contains a high level of Pro, when these two are mixed, the hydroxyl group of Ser/Tyr will react with the carboxyl group of Pro, thus leading to the formation of esters. At the same time, hydrogen bonds are formed among the hydroxyl group of Ser/Tyr, the amino group of Pro, and the hydroxyl group of starch. This disrupts the double helices of SPV, reducing its ordered region and increasing its boiling resistance.

Table 6 shows vermicelli’s thermal properties and secondary structure with gliadin fractions. The peak denaturation temperature (Tp) and enthalpy (ΔH) of ω-gliadin fractions are higher than those of αβγ-gliadin fractions due to their higher molecular weight. The enthalpies (ΔH) of SPV-containing gliadin fractions are all higher than those of the control, suggesting that adding gliadin fractions enhances the ordering of amylopectin in SPV. The main secondary structure of gliadin fractions is β-sheet and the difference between the structure of ω-gliadin and αβγ-gliadin fractions is that more random coils are present in the former and greater amounts of α-helix are present in the latter. A secondary structural conformational transition from β-sheet to α-helix, β-turn, and random coils happens after the interaction of ω-gliadin and sweet potato starch, as shown in Table 6. Otherwise, the interaction of αβγ-gliadin and sweet potato starch leads to the loss of its α-helix and β-turn structure. The secondary structure of gliadin fractions in the best SPV is as follows: 0.30% α-helix, 46.07% intermolecular β-sheet, 50.93% intramolecular β-sheet, 1.66% β-turn, and 1.04% random coils.

### 3.5. The 13C Solid-State NMR Spectra of Samples with Different Boiling Break Time of SPV

Figure 3 shows the *^13^C* solid-state NMR spectra of samples with different SPV boiling break times. According to reference [8,14,17], the resonances at 103.3/103.4 ppm, 82.8/82.7/83.0 ppm, 73.0/73.1/72.9/72.6 ppm, and 62.1 ppm in Figure 3 are, respectively, assigned to C1, C4, C2/C3/C5, and C6 sweet potato starches. Other resonances, shown in Figure 3i,j are assigned as follows: 173.5 ppm for the backbone CO of ω-/αβγ-gliadin, 157.6/157.0 ppm for the Tyr ζ (Y_ζ_) of ω-/αβγ-gliadin, 129.8/129.2 ppm for the Tyr γ, δ (Y_γ, δ_)/Phe C_ε_ (F_ε_) of ω-/αβγ-gliadin, 116.4 ppm for Tyr C_ε_ (Y_ε_), 56.6 ppm for Pro C_α_ (P_α_)/Thr C_α_ (T_α_)/Ser C_β_ (S_β_), 53.6 ppm for Gln C_α_ (Q_α_)/Leu C_α_ (L_α_), 48.9 ppm for Pro C_δ_ (P_δ_) 30.6 ppm for Gln γ (Q_γ_)/Pro β (P_β_), and 25.9 ppm for the Pro γ (P_γ_)/Leu γ (L_γ_) of ω-/αβγ-gliadin fractions.

Compared with Figure 3a,b, the resonances for C1 and C2/3/5 of SPV move slightly towards a higher wavenumber. This shift is a result of the swelling that occurs at high temperatures. It indicates that there is a greater amount of amylopectin in the sample shown in Figure 3b. The increased presence of amylopectin enhances the boiling resistance of vermicelli. By making a comparison between Figure 3c,d, it can be seen that the resonance of the vermicelli for C6 to a lower wavenumber after the addition of ω-gliadin. This shift indicates that the cause of the enhancement in the water resistance of the vermicelli is the formation of a hydrogen bond. This hydrogen bond is formed between the Tyr/Ser hydroxyl group in ω-gliadin (which contains more Tyr/Ser than αβγ-gliadin) and the C6 hydroxyl group of the vermicelli. The macromolecular polymers formed by ω-gliadin and sweet potato starch are very stable in structure. When comparing Figure 3e with Figure 3f, it is obvious that the chemical shift of vermicelli remains unchanged after the addition of αβγ-gliadin. It is possible that the improvement in the boiling resistance of vermicelli is related to its influence on the state of the granules. When comparing Figure 3g with Figure 3h, after adding the low-concentration ω-gliadin + αβγ-gliadin (1:1) mixture, as shown in Figure 3g, the chemical shift of C4 of SPV shifts slightly towards the higher wavenumber. This shift indicates that hydrogen bonds might be formed between the hydroxyl group in the gliadin fractions and C4 of SPV at this concentration.

The formation of these hydrogen bonds is not conducive to improving vermicelli’s boiling properties. The reason for this is that the addition of a high-concentration mixture can enhance the boiling resistance of vermicelli noodles, which might be related to the more stable structure of the protein–starch conjugate. This enhanced stability arises from the formation of a hydrogen bond between the hydroxyl group or amino group of the gliadin fractions and the C6 hydroxyl group of SPV. Comparing Figure 3i,j, it can be deduced that the resonance at 116.4 ppm in Figure 3i is a result of the Cε of Tyr because ω-gliadin contains a relatively higher amount of Tyr (although the specific data regarding this are not presented). The resonance around ~130 ppm in Figure 3i,j should be that of Phe. Given that αβγ-gliadin, shown in Figure 3j, contains a higher content of Phe (although the specific data are not shown), its resonance intensity is greater than that of ω-gliadin, shown in Figure 3i.

### 3.6. X-Ray Diffraction of Samples with Different Boiling Break Time of SPV

The XRD patterns of samples with different boiling break times of SPV are shown in Figure 4. The typical XRD patterns of ω-gliadin and αβγ-gliadin aggregates are first detected with diffraction angles at 2θ 17.38° and 9.78°/19.90° in Figure 4i,j, respectively. Their peak pattern is close to that reported in the literature [8], but there is no diffraction peak of sodium chloride and its purity is higher. According to the literature [27], the diffraction angles of sweet potato starch are 2θ 15°, 17°, 18° and 23°, but only one angle at 2θ ~13.7° is left in Figure 4a–c,e,g. The corresponding crystals should be ordered like amylopectin [14] in unbroken granules of dough binding; these granules are too small to be broken during the boiling process. The shorter boiling break time of these samples indicates that the presence of these crystals in small granules is the main reason for the reduced resistance of vermicelli to boiling. When ω-gliadin (Figure 4d) and αβγ-gliadin (Figure 4f) are added into SPV, its boiling break time increases and the crystal diffraction peak of SPV disappears. For ω-gliadin, a peak at 2θ 17.36° is present in the former sample, while new crystal diffraction peaks at 2θ 17.56°/19.84° appear in the latter sample. However, when both gliadin fractions are mixed into the vermicelli, the resistance of the vermicelli to water is not strong at low concentrations. The diffraction peak of amylopectin crystal is still present in the vermicelli, as shown in Figure 4g. As the content of gliadin complex content increases from 1% to 2% (Figure 4h), a diffraction peak of the ω-gliadin crystal is present. The crystal diffraction peak for amylopectin disappears, and the longest boiling break time is reached in the sample. The reason for this could be twofold: One is that ω-gliadin and αβγ-gliadin bind to the surface of unbroken sweet potato granules and block the pores, hindering the removal of hot water vapor in the granules during the heating process, leading to the breakage of the granules and the destruction of the internal amylopectin crystals. The second is that the ω-gliadin-containing α-helix structure is automatically arranged into an ordered crystal structure and slows the water absorption of SPV during the boiling process, thus extending the boiling break time.

### 3.7. A Schematic Diagram of Gliadin Fractions Addition on the Increase of Boiling Break Time of SPV

Figure 5 shows a schematic diagram of the interaction between gliadin fractions and sweet potato starch in vermicelli.

When ω-gliadin and αβγ-gliadin fractions are mixed, esterification occurs between the hydroxyl group of tyrosine in the former and the carboxyl group of proline in the latter, by which these two proteins are bound together (① in Figure 5). The etherification reaction occurs between the Ser hydroxyl group of ω-gliadin and the C6 hydroxyl group of starch during dehydration (② in Figure 5). Meanwhile, a hydrogen bond is formed between the amino group of Pro in the binding gliadin complex and another C6 hydroxyl group of sweet potato starches (③ in Figure 5), which causes the binding gliadin complex to absorb onto the surface of granules. As the heating process continues, the volume of water vapor within the granules bound to the gliadin fractions increases, leading the granules rupture as the gliadin membrane acts as a barrier to prevent the vapor from evaporating. After rupture, a hydrogen bond is formed between the amino group of Pro in the gliadin complex and the C6 hydroxyl group of the sweet potato amylopectin molecule (④ in Figure 5), which impedes the orderly arrangement of starch molecules for crystallization during refrigeration, thereby enhancing vermicelli’s resistance to boiling, increasing the encapsulation of gluten networks by starch particles [21]. This encapsulation causes more granules in SPV to break during the boiling process, and thus more amylose and amylopectin leach out to form a starch gel with better water resistance. Meanwhile, the crystal structure of granules that is not beneficial to the boiling resistance of vermicelli is also destroyed in this process. The resistance of SPV to boiling is further enhanced through the formation of crystals made of mixtures of the gliadin, the leached amylose, and amylopectin.

## 4. Conclusions

The longest boiling-resistant time of fresh SPV extends to 34.31 min by swelling the dough binder at 50 °C for 5 h, adding a 2% complex of ω-gliadin + αβγ-gliadin at a ratio of 1:1, and mixing at 70 °C for 20 min. This is 95.7% higher than the control, which has a time of 17.78 min. The reasons for the extended time include: the high content of amylopectin leached from granules via the swelling + freeze–thaw process is beneficial for improving the boiling resistance of SPV. Esterification between ω-gliadin and αβγ-gliadin fractions in SPV facilitates the formation of eutectic crystals between gliadin fractions and sweet potato amylopectin, thereby enhancing the boiling resistance of sweet potato vermicelli during the mixing process. The boiling resistance of vermicelli is positively correlated with the amorphous structure of the starch. The method of swelling + freeze–thaw with the addition of gliadin fractions is a practical way to produce high-quality SPV free from alum.

## Figures and Tables

**Figure 1 foods-14-00081-f001:**
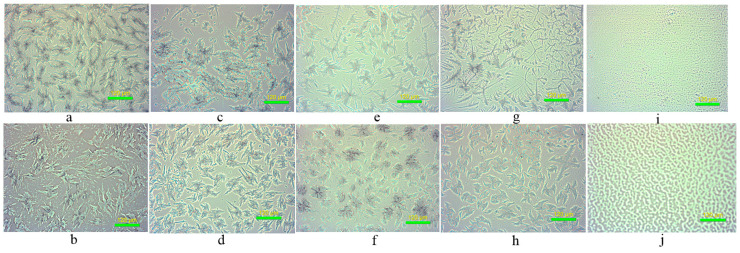
The light micrograph of different SPV. (**a**) SPV with shortest boiling break time (swollen at 40 °C for 2 h); (**b**) SPV with longest boiling break time (swollen at 50 °C for 5 h); (**c**) SPV for control 1 (stirred at 70 °C for 40 min); (**d**) SPV for control 1 + ω-gliadin; (**e**) SPV for control 2 (stirred at 40 °C for 60 min); (**f**) SPV for control 2 + αβγ-gliadin; (**g**) SPV with both gliadin fractions (shortest boiling break time); (**h**) SPV with both gliadin fractions (longest boiling break time); (**i**) ω-gliadin; (**j**) αβγ-gliadin.

**Figure 2 foods-14-00081-f002:**
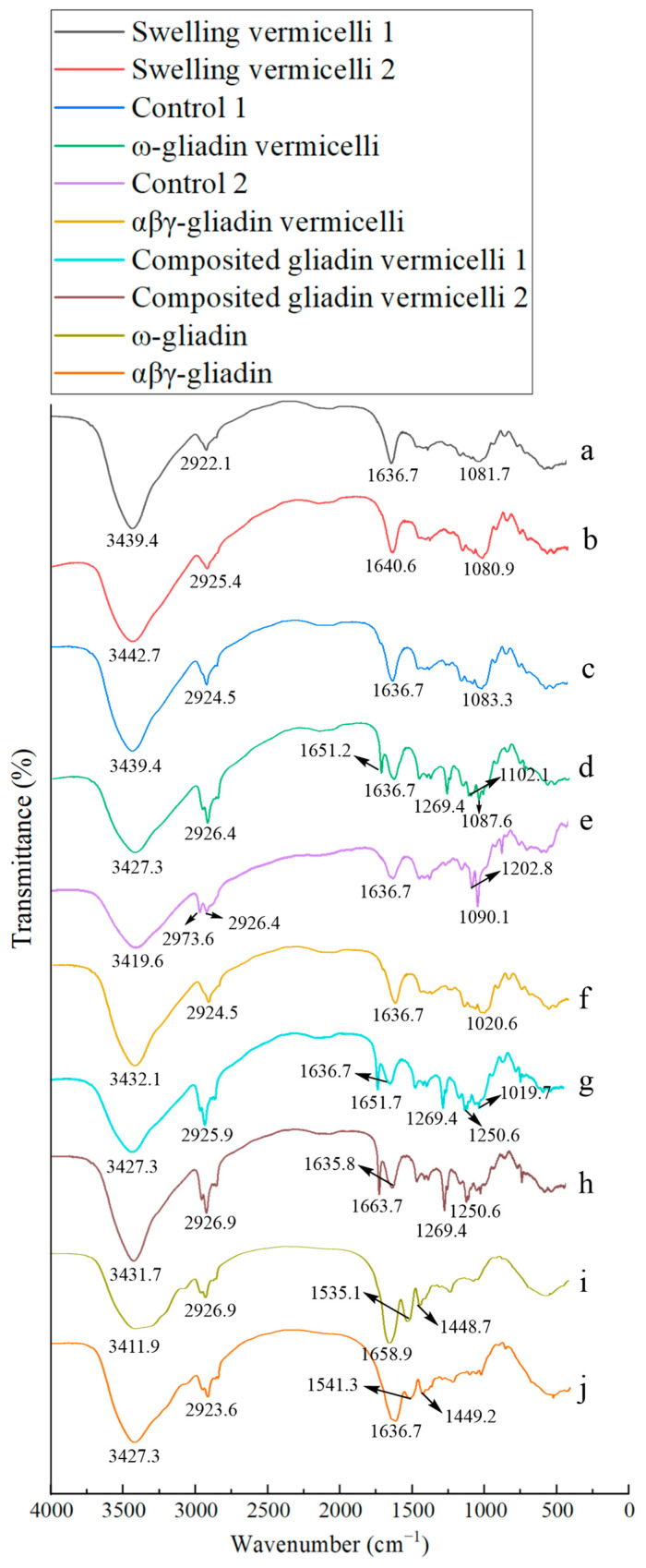
FTIR spectra of different SPV samples. Note: the sample labeling is the same as in Figure 1. (**a**) SPV with shortest boiling break time (swollen at 40 °C for 2 h); (**b**) SPV with longest boiling break time (swollen at 50 °C for 5 h); (**c**) SPV for control 1 (stirred at 70 °C for 40 min); (**d**) SPV for control 1 + ω-gliadin; (**e**) SPV for control 2 (stirred at 40 °C for 60 min); (**f**) SPV for control 2 + αβγ-gliadin; (**g**) SPV with both gliadin fractions (shortest boiling break time); (**h**) SPV with both gliadin fractions (longest boiling break time); (**i**) ω-gliadin; (**j**) αβγ-gliadin.

**Figure 3 foods-14-00081-f003:**
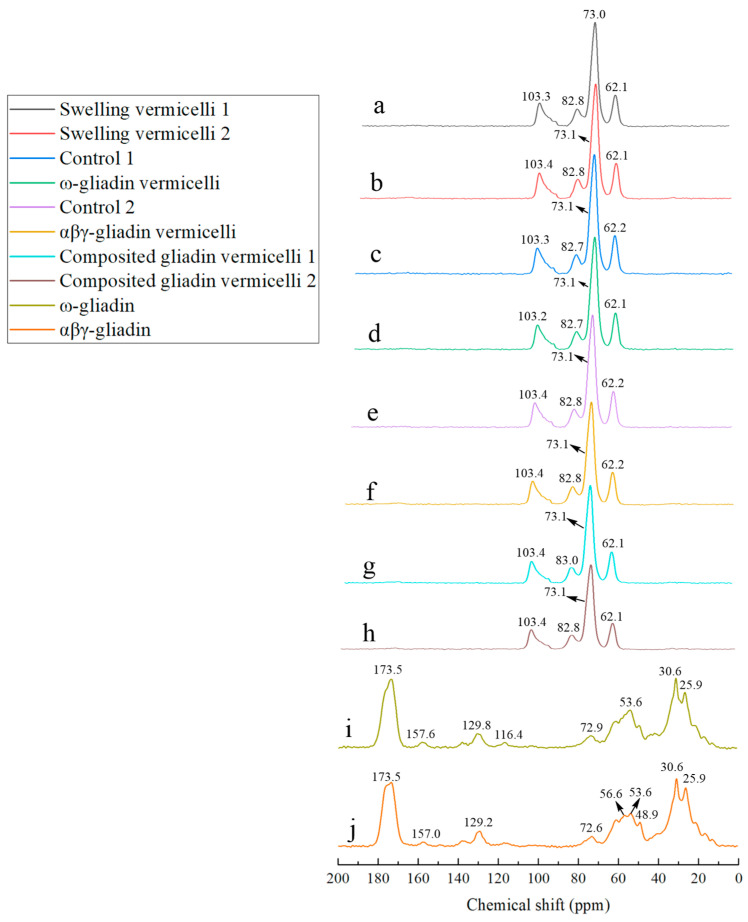
The *^13^C* solid-state NMR spectra of different SPV. Note: Sample labeling is the same as in Figure 1. (**a**) SPV with shortest boiling break time (swollen at 40 °C for 2 h); (**b**) SPV with longest boiling break time (swollen at 50 °C for 5 h); (**c**) SPV for control 1 (stirred at 70 °C for 40 min); (**d**) SPV for control 1 + ω-gliadin; (**e**) SPV for control 2 (stirred at 40 °C for 60 min); (**f**) SPV for control 2 + αβγ-gliadin; (**g**) SPV with both gliadin fractions (shortest boiling break time); (**h**) SPV with both gliadin fractions (longest boiling break time); (**i**) ω-gliadin; (**j**) αβγ-gliadin.

**Figure 4 foods-14-00081-f004:**
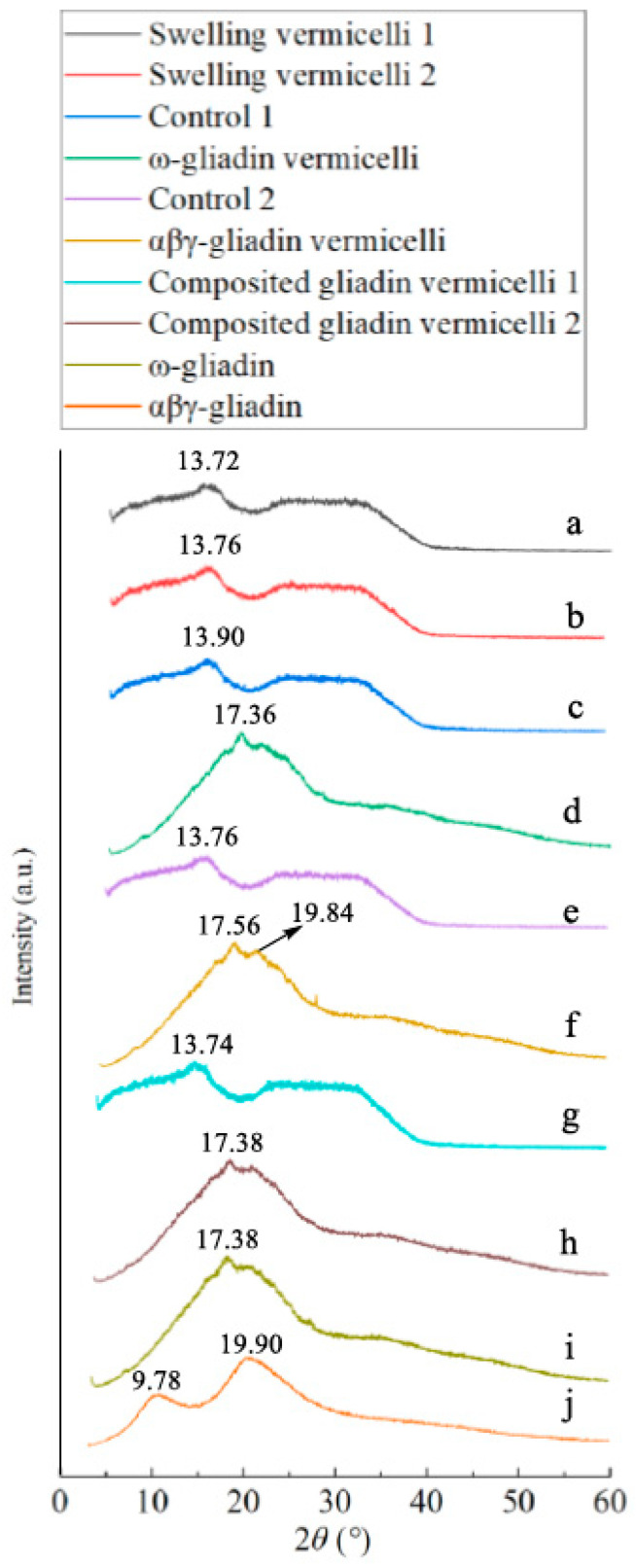
The X-ray diffraction of different SPV samples. Note: sample labeling is the same as in Figure 1. (**a**) SPV with shortest boiling break time (swollen at 40 °C for 2 h); (**b**) SPV with longest boiling break time (swollen at 50 °C for 5 h); (**c**) SPV for control 1 (stirred at 70 °C for 40 min); (**d**) SPV for control 1 + ω-gliadin; (**e**) SPV for control 2 (stirred at 40 °C for 60 min); (**f**) SPV for control 2 + αβγ-gliadin; (**g**) SPV with both gliadin fractions (shortest boiling break time); (**h**) SPV with both gliadin fractions (longest boiling break time); (**i**) ω-gliadin; (**j**) αβγ-gliadin.

**Figure 5 foods-14-00081-f005:**
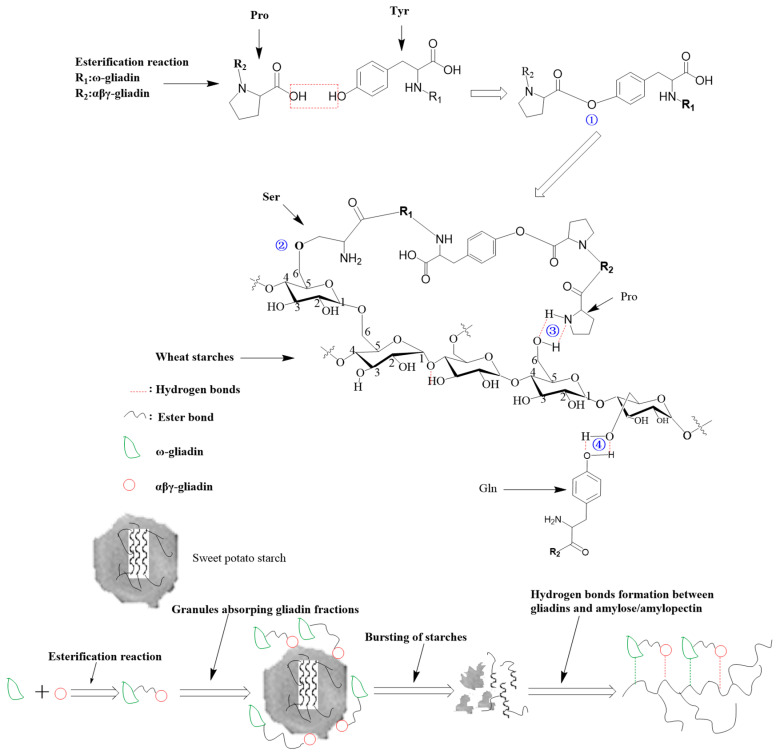
A schematic diagram of the interaction between gliadin fractions and sweet potato starches in vermicelli.

**Table 1 foods-14-00081-t001:** The effect of swelling temperature and time on the break time in boiling water (min).

Swelling Temperature/°C	Swelling Time/h				
	2	3	4	5	6
30 (poor formability)	3.83 ± 0.10	4.65 ± 0.18	3.94 ± 0.02	4.13 ± 0.04	4.58 ± 0.04
35 (poor formability)	4.65 ± 0.13	7.77 ± 0.22	8.20 ± 0.15	7.29 ± 0.39	10.89 ± 0.14
40	6.85 ± 0.17	9.33 ± 0.05	10.22 ± 0.13	9.23 ± 0.13	11.73 ± 0.38
50	11.67 ± 0.03 **	11.88 ± 0.05 **	12.53 ± 0.35 *	15.84 ± 0.34 **	15.13 ± 0.30 *
60	9.27 ± 0.05 **	9.01 ± 0.13	10.63 ± 0.26	12.06 ± 0.49 *	10.54 ± 0.24

Note: We observe poor formability. The vermicelli becomes fractured following the boiling process. * *p* ˂ 0.05, ** *p* ˂ 0.01.

**Table 2 foods-14-00081-t002:** The effect of the amount of ω-gliadin added, the mixing temperature, and the mixing time on the resistance of vermicelli to boiling (min).

ω-Gliadin Addition/%	Mixing Temperature/°C	Mixing Time/min		
		20	40	60
0	30	16.54 ± 0.54	15.77 ± 0.13	17.35 ± 0.02
	40	18.65 ± 0.03 *	18.64 ± 0.13 *	18.55 ± 0.20 *
	50	18.89 ± 0.26 *	18.45 ± 0.03 *	18.36 ± 0.18 *
	60	18.00 ± 0.22 *	18.51 ± 0.03 *	18.55 ± 0.03 *
	70	17.53 ± 0.08 *	17.78 ± 0.13 *	17.52 ± 0.02 *
0.1	30	21.44 ± 0.34 **	22.30 ± 0.57 *	24.35 ± 0.40 **
	40	23.92 ± 0.47 **	22.33 ± 0.03 **	21.54 ± 0.16 **
	50	23.16 ± 0.03 **	24.6 ± 0.28 **	22.50 ± 0.33 **
	60	21.09 ± 0.59 *	21.37 ± 0.25 **	23.10 ± 0.22 **
	70	24.58 ± 0.20 **	25.00 ± 0.42 **	23.20 ± 0.42 **
0.5	30	20.42 ± 0.15 **	20.86 ± 0.18 **	22.18 ± 0.28 **
	40	20.63 ± 0.10 **	20.91 ± 0.34 **	20.63 ± 0.26 **
	50	22.39 ± 0.28 **	21.11 ± 0.18 **	22.03 ± 0.62 *
	60	22.17 ± 0.12 **	21.87 ± 0.10 **	20.05 ± 0.27 *
	70	19.60 ± 0.05 **	20.23 ± 0.17 **	19.85 ± 0.17 **
0.8	30	23.10 ± 0.22 **	23.37 ± 0.23 **	24.43 ± 0.12 **
	40	23.02 ± 0.60 **	22.78 ± 0.09 **	20.88 ± 0.86 *
	50	22.82 ± 0.30 **	21.64 ± 0.04 **	22.45 ± 0.27 **
	60	20.49 ± 0.28 **	20.75 ± 0.20 **	20.86 ± 0.24 **
	70	21.83 ± 0.18 **	21.34 ± 0.13 **	20.44 ± 0.16 **
1	30	18.96 ± 0.21 *	20.05 ± 0.35 *	22.11 ± 0.28 **
	40	22.44 ± 0.24 **	22.71 ± 0.24 **	22.22 ± 0.02 **
	50	19.54 ± 0.16 *	20.70 ± 0.12 **	20.59 ± 0.28 **
	60	20.53 ± 0.34 *	20.67 ± 0.15 **	21.89 ± 0.26 **
	70	19.64 ± 0.06 **	19.01 ± 0.29 *	18.50 ± 0.15 *

* *p* ˂ 0.05, ** *p* ˂ 0.01.

**Table 3 foods-14-00081-t003:** The effect of the amount of αβγ-gliadin addition, mixing temperature, and mixing time on resistance of vermicelli to boiling (min).

αβγ-Gliadin Addition/%	Mixing Temperature/°C	Mixing Time/min		
		20	40	60
0.5	30	22.85 ± 0.13 **	23.18 ± 0.04 **	22.12 ± 0.35 **
	40	24.68 ± 0.80 **	22.63 ± 0.58 **	26.49 ± 0.23 **
	50	23.18 ± 0.20 **	25.90 ± 0.43 **	22.10 ± 0.23 **
	60	21.64 ± 0.44 **	23.56 ± 0.19 **	25.47 ± 0.57 **
	70	24.26 ± 0.16 **	24.66 ± 0.06 **	24.85 ± 0.42 **
1.0	30	27.23 ± 0.28 **	27.43 ± 0.12 **	28.83 ± 0.39 **
	40	27.98 ± 0.36 **	26.09 ± 0.17 **	24.78 ± 0.44 **
	50	23.65 ± 0.32 **	27.05 ± 0.17 **	26.59 ± 0.03 **
	60	26.54 ± 0.38 **	25.88 ± 0.13 **	25.58 ± 0.07 **
	70	23.32 ± 0.45 **	22.45 ± 0.20 **	23.79 ± 0.34 **
2.0	30	26.63 ± 0.27 **	27.44 ± 0.21 **	27.03 ± 0.32 **
	40	26.63 ± 0.31 **	26.72 ± 0.13 **	29.53 ± 0.18 **
	50	27.80 ± 0.37 **	26.04 ± 0.33 **	25.18 ± 0.34 **
	60	25.30 ± 0.12 **	26.37 ± 0.47 **	25.16 ± 0.34 **
	70	26.65 ± 0.13 **	27.43 ± 0.07 **	26.19 ± 0.32 **
4.0	30	28.16 ± 0.01 **	28.31 ± 0.34 **	26.57 ± 0.17 **
	40	26.56 ± 0.36 **	25.38 ± 0.16 **	25.78 ± 0.09 **
	50	25.55 ± 0.03 **	25.93 ± 0.06 **	24.04 ± 0.28 **
	60	24.97 ± 0.35 **	23.71 ± 0.01 **	24.73 ± 0.13 **
	70	25.13 ± 0.32 **	25.73 ± 0.19 **	24.83 ± 0.13 **

** *p* ˂ 0.01.

**Table 4 foods-14-00081-t004:** The effect of the amount of ω-gliadin and αβγ-gliadin addition at a ratio of 1:1, mixing temperature, and mixing time with starch on the resistance of vermicelli to boiling (min).

ω + αβγ-Gliadin Addition/%	Mixing Temperature/°C	Mixing Time/min		
		20	40	60
0.5	30	22.83 ± 0.20 **	23.28 ± 0.23 **	22.95 ± 0.25 **
	40	21.58 ± 0.10 **	21.85 ± 0.15 **	21.61 ± 0.06 **
	50	21.77 ± 0.12 **	22.21 ± 0.29 **	21.33 ± 0.13 **
	60	22.45 ± 0.10 **	23.03 ± 0.28 **	21.31 ± 0.18 **
	70	24.11 ± 0.31 **	23.83 ± 0.18 **	25.38 ± 0.27 **
1.0	30	20.30 ± 0.07 **	24.98 ± 0.22 **	24.43 ± 0.11 **
	40	20.25 ± 0.13 **	21.40 ± 0.22 **	23.07 ± 0.27 **
	50	23.41 ± 0.09 **	24.73 ± 0.56 **	27.08 ± 0.13 **
	60	22.82 ± 0.22 **	26.99 ± 0.23 **	26.38 ± 0.47 **
	70	21.92 ± 0.37 **	22.28 ± 0.33 **	22.33 ± 0.38 **
2.0	30	24.52 ± 0.00 **	22.12 ± 0.28 **	23.07 ± 0.10 **
	40	22.61 ± 0.14 **	22.14 ± 0.28 **	22.00 ± 0.08 **
	50	21.25 ± 0.15 **	21.06 ± 0.07 **	21.73 ± 0.24 **
	60	25.52 ± 0.22 **	32.89 ± 0.66 **	29.77 ± 0.60 **
	70	34.31 ± 0.21 **	32.25 ± 0.10 **	28.53 ± 0.14 **
4.0	30	24.43 ± 0.18 **	24.15 ± 0.03 **	27.52 ± 0.22 **
	40	23.20 ± 0.17 **	20.30 ± 0.47 *	24.90 ± 0.47 **
	50	24.01 ± 0.18 **	25.44 ± 0.13 **	30.68 ± 0.27 **
	60	23.72 ± 0.20 **	21.43 ± 0.20 **	20.56 ± 0.49 *
	70	32.71 ± 0.13 **	21.55 ± 0.20 **	21.73 ± 0.05 **

* *p* ˂ 0.05, ** *p* ˂ 0.01.

**Table 5 foods-14-00081-t005:** Effects of cold storage time on resistance of vermicelli to boiling (min).

Cooking Time/min	Storage Time/d									
	1	2	3	4	5	6	7	8	9	10
Swelling vermicelli 1	5.0 ± 0.3	10.9 ± 0.2 *	13.7 ± 0.3	12.8 ± 1.1 *	24.1 ± 1.3 *	23.3 ± 0.4	23.8 ± 2.3	24.6 ± 0.1	25.6 ± 2.1	19.5 ± 1.0 *
Swelling vermicelli 2	8.7 ± 0.5	8.9 ± 3.7	10.6 ± 4.2	19.2 ± 1.4 *	34.0 ± 0.7	29.0 ± 1.7	34.2 ± 1.7	34.3 ± 0.9	33.9 ± 0.6 *	23.7 ± 0.8 **
Control 1	7.2 ± 0.2	24.2 ± 1.6	27.7 ± 0.8	27.7 ± 3.8	22.7 ± 1.5	30.0 ± 0.6	27.7 ± 0.0	33.6 ± 0.8	30.1 ± 0.4	30.6 ± 0.3 **
ω-gliadin vermicelli	4.3 ± 0.2	10.1 ± 0.0 *	13.3 ± 0.3	16.2 ± 0.9 *	25.1 ± 1.2	26.9 ± 1.6	25.8 ± 1.0	29.6 ± 0.6	34.3 ± 0.9	33.9 ± 0.6 **
Control 2	13.7 ± 0.3 *	18.7 ± 1.5	33.0 ± 1.6	38.6 ± 0.8	39.5 ± 1.1	32.9 ± 0.5	22.3 ± 4.98	26.6 ± 0.8 *	35.9 ± 0.7	36.8 ± 0.6 **
αβγ-gliadin vermicelli	9.7 ± 1.0	17.4 ± 0.2 *	15.6 ± 5.1	25.9 ± 0.8	27.6 ± 0.6	26.6 ± 0.6 *	30.3 ± 0.1 *	32.3 ± 0.1	32.6 ± 0.3	30.8 ± 0.3 **
Composited gliadin vermicelli 1	10.7 ± 0.6	13.7 ± 0.4	19.8 ± 1.8 *	37.9 ± 0.5	40.6 ± 0.6	38.8 ± 1.7	39.3 ± 1.3	37.8 ± 0.4	35.9 ± 0.4	32.2 ± 1.0 *
Composited gliadin vermicelli 2	15.2 ± 1.8	24.5 ± 0.7	24.5 ± 1.7	27.2 ± 0.4	28.4 ± 0.7	30.8 ± 1.2	27.0 ± 0.0 *	30.2 ± 0.6	28.6 ± 1.3	32.2 ± 1.2

Note: swelling vermicelli 1—40 °C for 2 h; swelling vermicelli 2—50 °C for 5 h; xontrol 1—70 °C for 40 min without gliadin; ω-gliadin vermicelli—0.1% ω-gliadin; xontrol 2—40 °C for 60 min without gliadin; αβγ-gliadin vermicelli—2% αβγ-gliadin; composite gliadin 1 vermicelli—1% of ω-gliadin + αβγ-gliadin (1:1) (40 °C for 20 min); composited gliadin 2 vermicelli—2% of ω-gliadin + αβγ-gliadin (1:1) (70 °C for 20 min). * *p* ˂ 0.05, ** *p* ˂ 0.01.

**Table 6 foods-14-00081-t006:** The thermal properties and secondary structure of vermicelli with gliadin fractions (%).

Samples	Thermal Properties		Secondary Structure				
	Tp1	ΔH1	α-Helix Content	Intermolecular β-Sheet Content	Intra-Molecular Aggregation Extended β-Sheet Content	β-Turn Content	Random Coils Content
Swelling vermicelli 1	99.72 ± 0.41	258.43 ± 1.45					
Swelling vermicelli 2	100.05 ± 0.25	254.67 ± 0.69					
Control 1	100.59 ± 0.57	232.7 ± 2.80					
ω-gliadin vermicelli	98.07 ± 0.07	260.82 ± 1.03 *	39.54	3.85	15.17	25.77	15.67
Control 2	100.27 ± 0.03	240.78 ± 0.50					
αβγ-gliadin vermicelli	98.95 ± 0.01 **	268.79 ± 0.22 **	0	53.71	46.29	0	0
Composited gliadin vermicelli 1	98.72 ± 0.10	269.59 ± 6.98	0	52.41	47.59	0	0
Composited gliadin vermicelli 2	97.72 ± 0.17	285.44 ± 3.18	0.30	46.07	50.93	1.66	1.04
ω-gliadin	93.29 ± 0.42	142.35 ± 0.01	0.28	22.45	63.36	12.08	1.83
αβγ-gliadin	89.51 ± 0.16 *	137.36 ± 0.83 *	2.40	23.25	61.45	12.90	0

Note: Swelling vermicelli 1—40 °C for 2 h. Swelling vermicelli 2—50 °C for 5 h. Control 1—70 °C for 40 min without gliadin; ω-gliadin vermicelli—0.1% ω-gliadin. Control 2—40 °C for 60 min without gliadin; αβγ-gliadin vermicelli—2% αβγ-gliadin. Composited gliadin 1 vermicelli—1% of ω-gliadin + αβγ-gliadin (1:1) (40 °C for 20 min). Composited gliadin 2 vermicelli—2% of ω-gliadin + αβγ-gliadin (1:1) (70 °C for 20 min). * *p* ˂ 0.05, ** *p* ˂ 0.01.

## Data Availability

The original contributions presented in the study are included in the article, further inquiries can be directed to the corresponding authors.

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
