# Peer review of "Preparation and Mechanism Analysis of Boiling Resistance of the Fresh Alum-Free Sweet Potato Vermicelli Containing Gliadin Fractions"

_foods, 2025, doi:10.3390/foods14010081_

Round 1
Reviewer 1 Report
Comments and Suggestions for Authors
This is a good study with a lot of well-done technical work, to replace alum with gliadin fractions in sweet potato vermicelli to avoid the harm of alum for the human body. Because of the viscosity of gliadins, the addition enhanced the boiling resistance of the product, as it was hypothesized by the authors.
Although there have been other treatments in this research field, apparently those did not supply the boiling resistance of the vermicelli as the added gliadin fractions.
The very long paragraphs, such as those in lines 185-212, 316-372, and 401-433, are difficult to read. Could you please rewrite these in two or three shorter paragraphs?
In Table 3 some colors are described but I can not see any.
Please use the wider format for Tables 5 and 6.
The legend in Figure 1 is difficult to follow. I think the format for the panel descriptions is so different (see authors guidelines).
Author Response
Respond to reviewer 1:
This is a good study with a lot of well-done technical work, to replace alum with gliadin fractions in sweet potato vermicelli to avoid the harm of alum for the human body. Because of the viscosity of gliadins, the addition enhanced the boiling resistance of the product, as it was hypothesized by the authors.
Although there have been other treatments in this research field, apparently those did not supply the boiling resistance of the vermicelli as the added gliadin fractions.
The very long paragraphs, such as those in lines 185-212, 316-372, and 401-433, are difficult to read. Could you please rewrite these in two or three shorter paragraphs?
Answer:These sentences have been modified by an expert in English.
In Table 3 some colors are described but I can not see any.
Answer:All colors have been highlighted in Table 1~5, they are marked as the shortest (green) and longest times of boiling resistance.
Please use the wider format for Tables 5 and 6.
Answer:The format for Table 5 and 6 has been enlarged.
The legend in Figure 1 is difficult to follow. I think the format for the panel descriptions is so different (see authors guidelines).
Answer:The legend in Figure 1 has been revised.
Reviewer 2 Report
Comments and Suggestions for Authors
The subject of the manuscript is appropriate and original. It fits into the theme of the Foods journal. The authors' research provides innovative information on the possibility of eliminating alum, which is added to the production of thin pasta. The authors, based on scientific reports, believe that alum is harmful to health. Therefore, the research conducted by the authors seems justified. Due to the experiment conducted, this article is very original, has great scientific value, and the obtained results have a practical dimension and can be used in the food industry.
The abstract does not raise any objections. The content contains information about the experiment conducted.
Introduction:
The introduction presents the problem, based on good literature, but the authors should clearly present the purpose of at the end of the content in the "Introduction" chapter
Materials and methods
The research methods and methods of solutions are correctly presented. The manuscript explains how the material for the research was collected. The analytical devices are presented. The estimation methods are appropriately selected and do not raise any objections. The authors should provide the year of the research in subsection 2.1. Results and discussion is good
Conclusions
The conclusions are too general, they should include a summary of the research results obtained. Please complete this.
Please note a few minor errors:
Please remove the incorrect entry:
line 34, remove (2019)
line 34-35 to mg/Kg correct to mg/kg (in all entries)
line 47 remove (2020e)
In the body of the paper, the authors use a space between the number and ℃ and without a space (please standardize this)
Table headers without bold
Figure 3. Correct the record
Reference: Correct the literature records in accordance with the editorial requirements of the Foods journal.
Item 2 is sci correct Sci
correct the year in italics.
The work, after taking into account minor additions, can be developed in the Foods journal.
Comments on the Quality of English Language
Author Response
Respond to reviewer 2:
The subject of the manuscript is appropriate and original. It fits into the theme of the Foods journal. The authors' research provides innovative information on the possibility of eliminating alum, which is added to the production of thin pasta. The authors, based on scientific reports, believe that alum is harmful to health. Therefore, the research conducted by the authors seems justified. Due to the experiment conducted, this article is very original, has great scientific value, and the obtained results have a practical dimension and can be used in the food industry.
The abstract does not raise any objections. The content contains information about the experiment conducted.
Introduction:
The introduction presents the problem, based on good literature, but the authors should clearly present the purpose of at the end of the content in the "Introduction" chapter
Answer:The purpose of the paper has been added at the end of introduction chapter.
Materials and methods
The research methods and methods of solutions are correctly presented. The manuscript explains how the material for the research was collected. The analytical devices are presented. The estimation methods are appropriately selected and do not raise any objections. The authors should provide the year of the research in subsection 2.1. Results and discussion is good.
Answer:The year of the research in subsection 2.1 has been added.
Conclusions
The conclusions are too general, they should include a summary of the research results obtained. Please complete this.
Answer:The conclusions have been written.
Please note a few minor errors:
Please remove the incorrect entry:
line 34, remove (2019)
Answer:The error has been corrected.
line 34-35 to mg/Kg correct to mg/kg (in all entries)
Answer:The error has been corrected.
line 47 remove (2020e)
Answer:The error has been corrected.
In the body of the paper, the authors use a space between the number and ℃ and without a space (please standardize this)
Answer:The error has been corrected.
Table headers without bold
Answer:The errors have been corrected.
Figure 3. Correct the record
Answer:The error has been corrected.
Reference: Correct the literature records in accordance with the editorial requirements of the Foods journal.
Item 2 is sci correct Sci
Answer:The error has been corrected.
correct the year in italics.
Answer:The error has been corrected.